# Improved Padding in CNNs for Quantitative Susceptibility Mapping

## Abstract

Deep learning methods have been proposed for quantitative susceptibility mapping (QSM) - background field removal, single-step QSM, and field-to-source inversion. However, the conventional padding mechanism used in CNNs can cause spatial artifacts, especially at the boundaries of regions of interest. To address this issue, we propose an improved padding technique which utilizes the neighboring voxels to estimate the invalid pixels at volume boundaries. Studies using simulated data show that the proposed method greatly improves estimation accuracy and reduces artifacts in the results. The code is available at .

**Keywords:** Padding, QSM.

## 1. Introduction

In QSM, tissue susceptibility is quantitatively estimated by extracting Larmor frequency offsets from complex MR signals to solve for the source tissue susceptibility (Wang and Liu, 2015). QSM processing usually involves a series of post-processing procedures, including (1) estimating the magnetic field from the raw MR phase data, (2) eliminating the background field contributions from outside the region of interest (ROI) to determine the local field, (3) solving the field-to-source inverse problem. In single-step QSM, the tissue susceptibility is directly estimated from the total field without background field removal. Both background field removal and field-to-source inversion require to solve ill-posed inverse problems. With the development of deep learning (DL), recent efforts have demonstrated the advantages of DL for QSM in background field removal(Bollmann et al., 2019a; Liu and Koch, 2019), field-to-source inversion(Yoon et al., 2018; Bollmann et al., 2019b; Jung et al., 2020; Gao et al., 2021; Chen et al., 2019), and single-step QSM(Wei et al., 2019). All these methods utilized U-Net (Ronneberger et al., 2015) like architecture with convolutional layers, max-pooling layers, and deconvolutional layers etc. However, these methods failed to consider the invalid pixels outside of ROIs, which could introduce inaccurate learning close to volume boundaries and cause spatial artifacts in the final results. Recent studies have found that the padding mechanism can cause spatial artifacts in CNNs. Through investigation on conventional padding techniques such as zero-padding, symmetric padding, and reflective padding, we found that in CNNs for background field removal and single-step QSM, the strong background field at the boundaries further hinder the learning process.

To address this problem, a new padding mechanism was proposed. The padding mechanism uses the neighboring voxels of feature maps to estimate the invalid voxels at image boundaries. We used simulated data for quantitative evaluation on the tasks of background field removal, field-to-source inversion, and single-step QSM tasks.

## 2. Method

Let $X$ are the feature values (pixels values) and $M$ is the corresponding binary mask. First, a convolution with all-one 3x3x3 kernel was the padded binary mask to get the scaling factor $1/sum(M)$ which applies appropriate scaling to adjust for the varying amount of invalid inputs. Second, a convolution with all-one 3x3x3 kernel was the each feature map to get the average value of valid neighboring pixels for the invalid pixels. For better generalization, the convolution kernels for feature maps and binary mask were trainable, which were initialized with all-one. After each convolution operation, the mask was not updated.

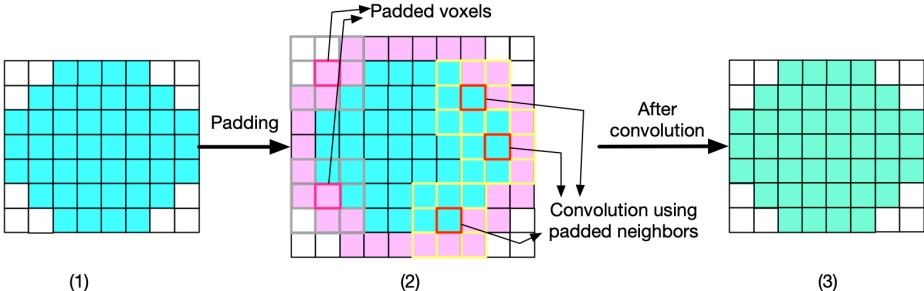

Figure 1: Illustration of Padding. In the feature map (1), the valid pixels show blue and invalid with white. In (2), the invalid pixels at the boundaries are estimated from its neighboring valid pixels, with color pink. In (3), the feature map at valid position are updated after convolution.

## 3. Experiments

We used the COSMOS result of 2016 QSM reconstruction challenge to generate simulated data. We applied random elastic transform, contrast change, and adding pseudo high susceptibility sources to augment the single QSM. The background field were simulated by using random placed background susceptibility sources with large susceptibility value outside the brain. The dipole convolution were then performed to get the induced field from the susceptibility distribution.

100 datasets with matrix size 160x160x160 and voxel size 1.0x1.0x1.0mm$^3$ were generated for network training tasks for background field removal, field-to-source inversion, and single-step. The network adopted a 3D U-Net like architecture, patch-based training with patch size 96x96x96, and L2 loss. We compared four padding mechanisms - zero padding, reflective padding, symmetric padding, and the proposed one. 100 testing datasets were generated using the same way as training data. The prediction results were evaluated with respect to the ground truth using quantitative metrics, peak signal-to-noise ratio (PSNR), normalized root mean squared error (NRMSE), high frequency error norm (HFEN), and structure similarity (SSIM) index.

## 4. Results

Table.1 displays the quantitative evaluation results. In the tasks of background field removal, field-to-source inversion, and single-step QSM, the proposed method achieved the best scores in all metrics.

Table 1: Quantitative evaluation on 100 synthetic testing data.

|  | PSNR (dB) | NRMSE (%) | HFEN (%) | SSIM (0-1) |
|---|---|---|---|---|
| **Background field removal** | | | | |
| zero padding | $50.3 \pm 5.3$ | $12.8 \pm 2.3$ | $11.5 \pm 2.6$ | $0.998 \pm 0.001$ |
| symmetric padding | $49.1 \pm 5.4$ | $14.7 \pm 2.3$ | $12.5 \pm 2.4$ | $0.998 \pm 0.002$ |
| reflective padding | $49.1 \pm 5.3$ | $14.7 \pm 2.4$ | $12.7 \pm 2.7$ | $0.998 \pm 0.002$ |
| neighbor padding | $\mathbf{52.6 \pm 5.2}$ | $\mathbf{9.9 \pm 1.6}$ | $\mathbf{8.9 \pm 1.8}$ | $0.999 \pm 0.001$ |
| | | | | |
| **Field-to-source Inversion** | | | | |
| zero padding | $45.3 \pm 4.2$ | $19.0 \pm 1.8$ | $19.2 \pm 1.5$ | $0.984 \pm 0.010$ |
| symmetric padding | $45.0 \pm 4.3$ | $19.6 \pm 1.8$ | $19.9 \pm 1.6$ | $0.984 \pm 0.010$ |
| reflective padding | $44.7 \pm 4.3$ | $20.3 \pm 1.8$ | $20.8 \pm 1.7$ | $0.983 \pm 0.010$ |
| neighbor padding | $\mathbf{46.0 \pm 4.2}$ | $\mathbf{17.4.2 \pm 1.7}$ | $\mathbf{17.3 \pm 1.2}$ | $0.986 \pm 0.009$ |
| | | | | |
| **Single-step QSM** | | | | |
| zero padding | $42.6 \pm 4.4$ | $25.8 \pm 2.2$ | $27.7 \pm 2.7$ | $0.974 \pm 0.016$ |
| symmetric padding | $42.3 \pm 0.6$ | $26.8 \pm 2.8$ | $28.9 \pm 3.0$ | $0.973 \pm 0.017$ |
| reflective padding | $42.1 \pm 4.5$ | $27.4 \pm 2.4$ | $29.2 \pm 2.9$ | $0.972 \pm 0.017$ |
| neighbor padding | $\mathbf{44.6 \pm 4.3}$ | $\mathbf{20.6 \pm 1.9}$ | $\mathbf{21.4 \pm 1.9}$ | $0.983 \pm 0.010$ |

## 5. Discussion and Conclusion

The proposed padding demonstrated better performance than other three padding mechanism in all three deep learning tasks for QSM. In the tasks of background field and single-step QSM, the proposed methods significantly outperformed and showed substantial less error. This may be due to that the strong background field contamination close to brain boundaries (tissue air interface) causes the conventional padding mechanism inefficient.

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

## Appendix A.

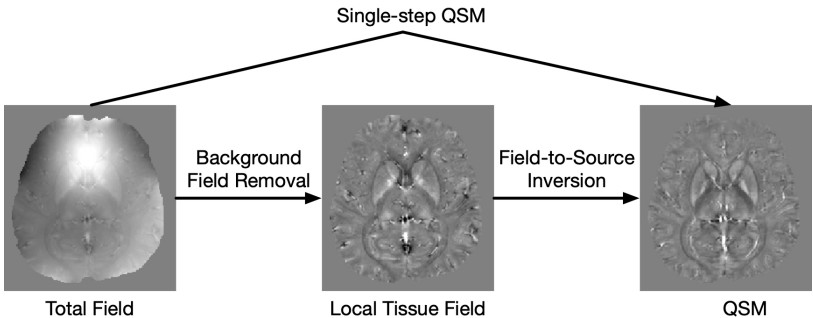

Figure 2: Illustration of QSM processing.

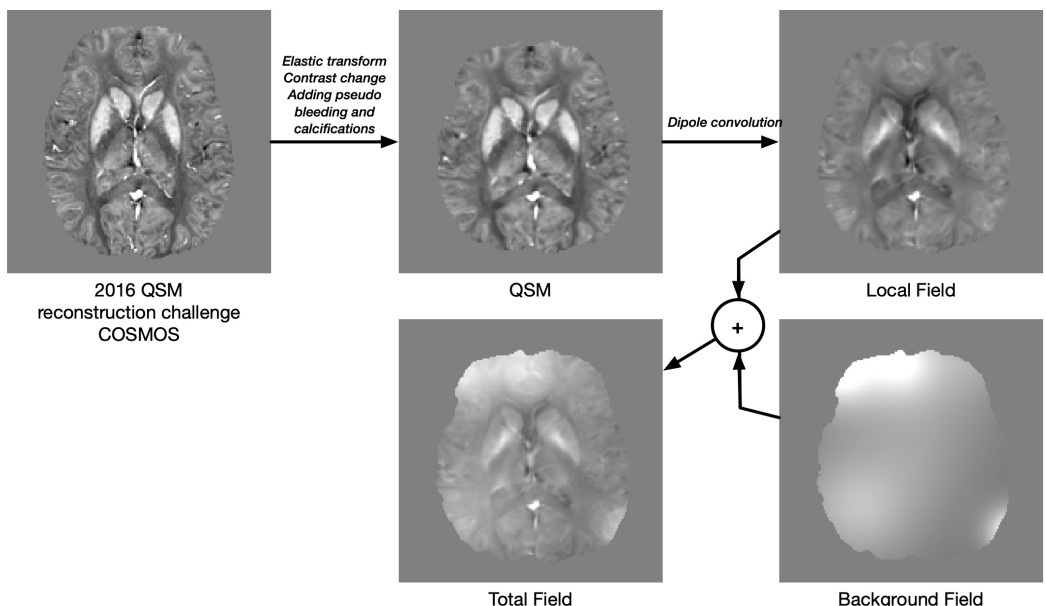

Figure 3: Illustration of simulation data.

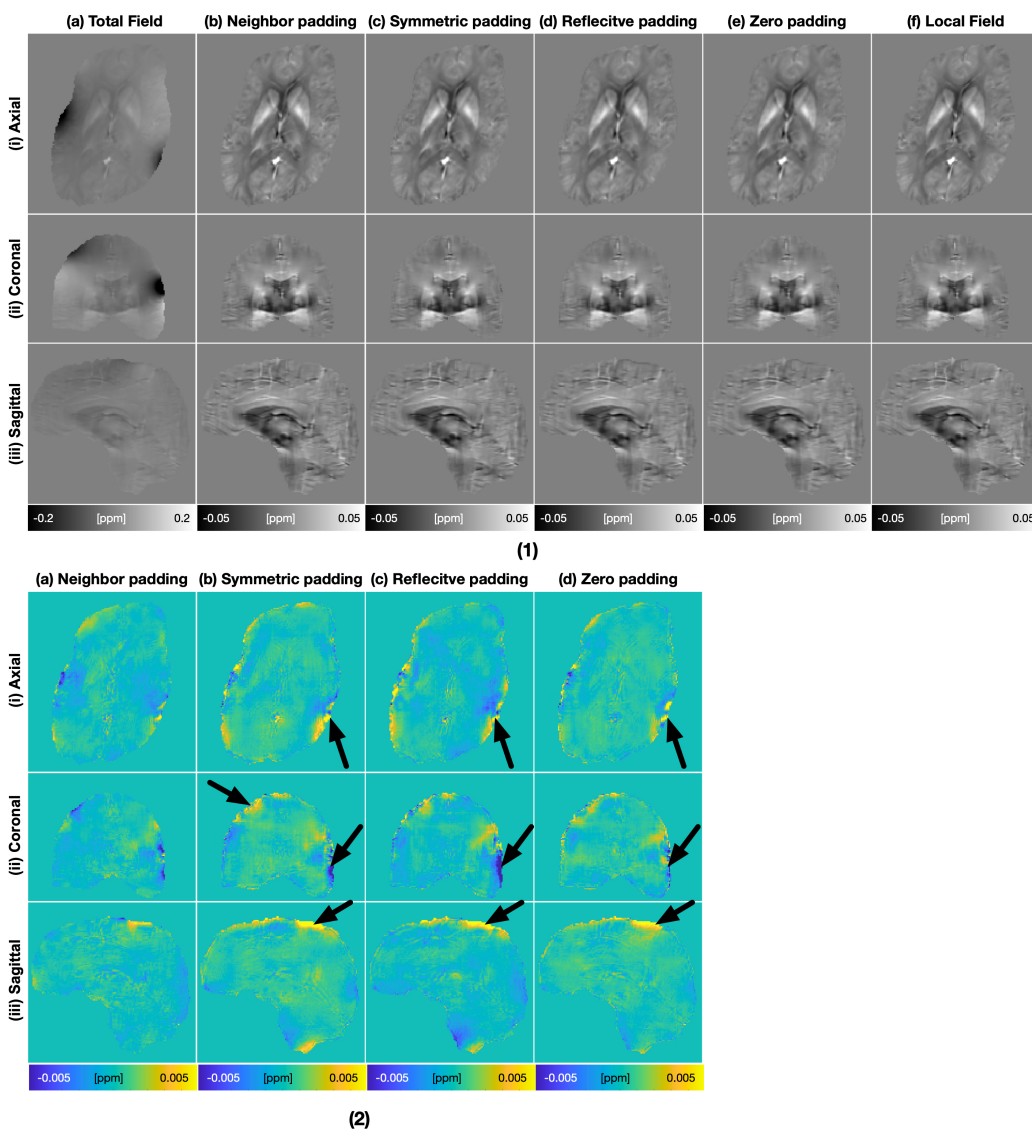

Figure 4: Comparison of background field removal performance on an example of testing data. From the residual error map (2), it is clearly showing that the proposed padding method has less residual errors, especially close to brain boundaries.

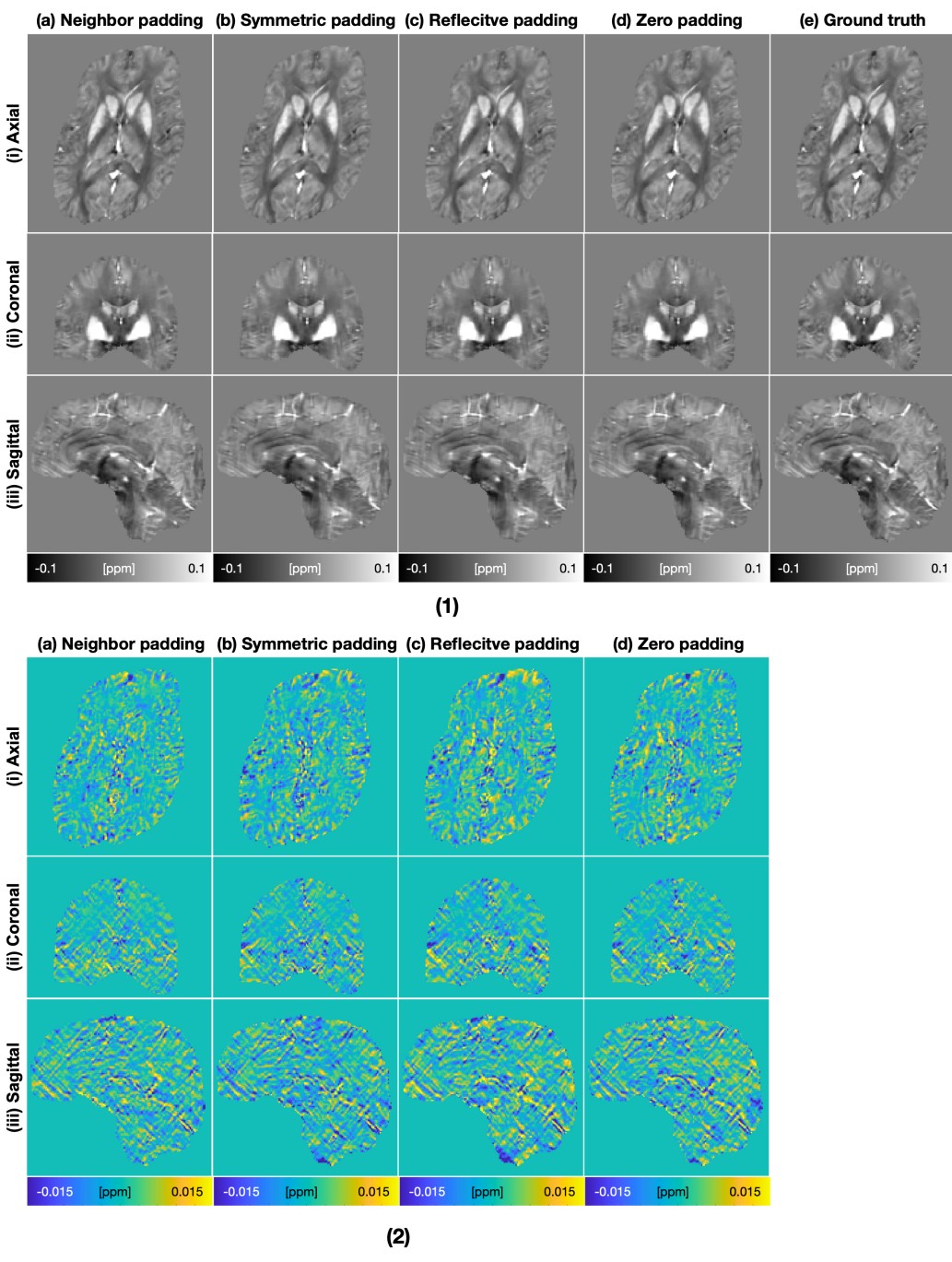

Figure 5: Comparison of field-to-source performance on an example of testing data. From the residual error map, all padding mechanisms shows comparable performance and the proposed padding method has less residual errors.

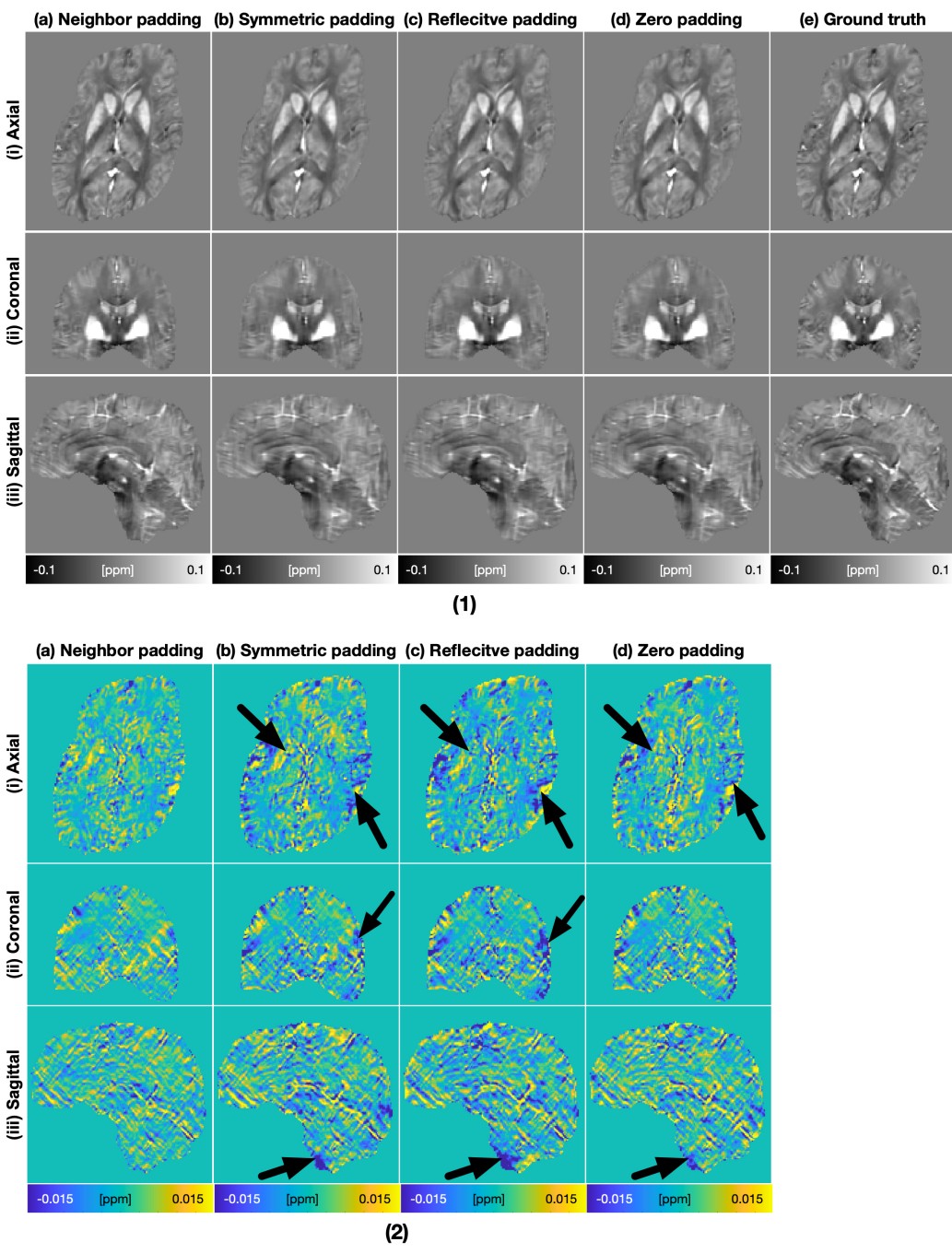

Figure 6: Comparison of single-step QSM performance on an example of testing data. From the residual error map (2), it is clearly showing that the proposed padding method has less residual errors, especially close to brain boundaries.

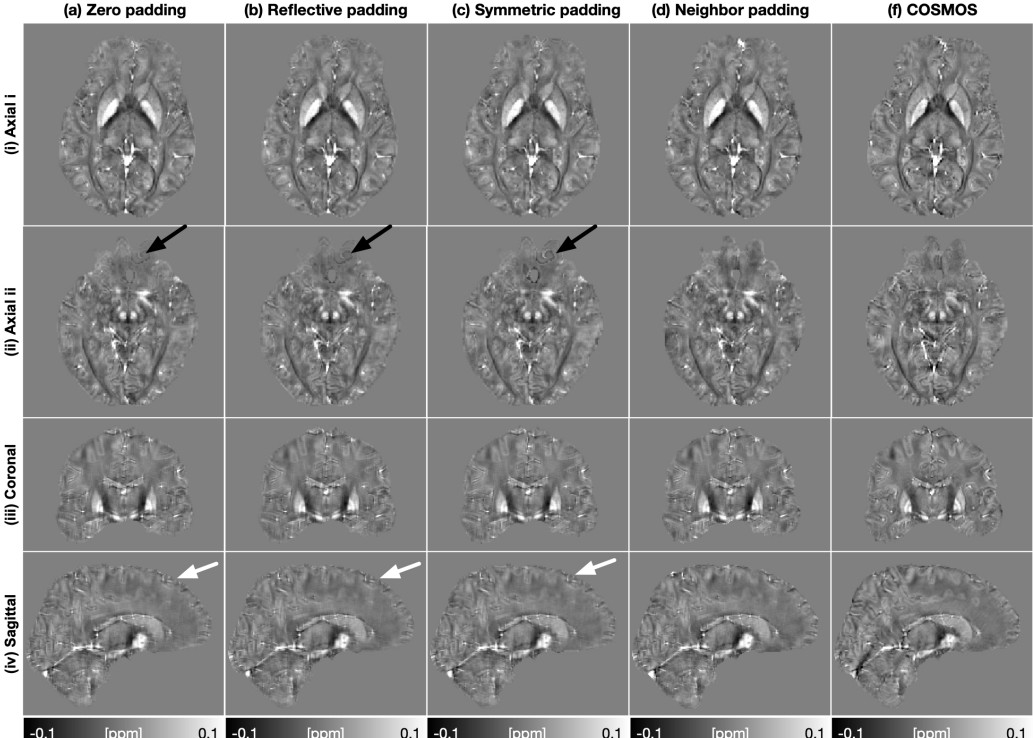

Figure 7: Comparison of single-step QSM performance on an in-vivo data. In the results of zero padding, reflective padding and symmetric padding show the obvious artifacts (ii, black arrows) and jagged-like artifacts at the boundaries (iv, white arrows).

