# OpenReview forum: "Improved Padding in CNNs for Quantitative Susceptibility Mapping"
_MIDL.io/2021/Conference/Short — Submitted to MIDL 2021_

### Official Review · Reviewer_ZYgn · 2021-04-23

**Confidence:** 3
**Final Rating:** 2

**Summary:**

The authors propose a new way to perform padding in CNNs to improve the performance on the quantitative susceptibility mapping task, due to an hypothesis that the conventional padding could be improved to cause less spatial artifacts. The study used simulated data. The authors showed quantitative and qualitative improvement with the new padding strategy.

**Strengths:**

The paper is overall well written with good figures and quantitative evaluation of the proposed method, consisting of essentially using convolutions for padding. The authors displayed that the proposed padding method improved results for the QSM task, with, in my opnion, valid reasoning as to why the traditional padding might worsen performance on QSM.



**Weaknesses:**

The explanation of the padding method, which is the main contribution of this manuscript, should be improved in Section 2. To me, it sounded confusing.

Short papers are strictly limited to 3 pages (including references), according to the MIDL submission guidelines.





**Deanonymize Review:**

yes

**Detailed Comments:**

"Recent studies have found that the padding mechanism can cause spatial artifacts in CNNs." lacks citation.

"This may be due to that the strong background field contamination close to brain
boundaries (tissue air interface) causes the conventional padding mechanism inefficient." This sentence could be rewritten as a better conclusion to the manuscript.

Figure 1 is overlapping with text from Section 2

There is no citation for the dataset, its not clear to the reader what "COSMOS result of 2016 QSM" is.

Code is not available in the submission, although it appears the authors intend to add it in an eventual camera ready submission.


**Justification Of The Rating:**

Short papers are strictly limited to 3 pages (including references).

The paper would need a rewrite to satisfy submission requirements, with minor writing improvements and summarization the paper could be reduced to 3 pages.

**Paper Type:**

methodological development

**Special Issue:**

no

---

### Official Review · Reviewer_JrLn · 2021-04-30

**Confidence:** 5
**Final Rating:** 4

**Summary:**

An interesting paper addressing boundary feature map issues for QSM inverse problem. The proposed padding strategy was validated on three different image-to-image mapping tasks in QSM with extensive experiments, which made it a solid paper. Such padding strategy may be generally applied to other tasks in medical imaging.

**Strengths:**

The proposed padding method was simple and effective. It worked well for three different image-to-image mapping tasks in QSM, which implies it may be applied to other medical image tasks using deep learning.

**Weaknesses:**

1. The code is not available. It will be a good contribution to share the source code for the community.
2. The content exceeds 3-page limit including reference. Please rearrange the paper to strictly follow the 3-page limit.

**Deanonymize Review:**

no

**Justification Of The Rating:**

The padding idea in this paper was proved to work on three image-to-image mapping tasks in QSM, with comprehensive experiments to evaluate the performance. The proposed method seems to be general for other tasks which will be interesting to be validated if possible.

**Paper Type:**

both

**Special Issue:**

no

---

### Meta-Review · Area_Chair_rUjb · 2021-05-07

**Recommendation:** Reject
**Confidence:** 5

**Metareview:**

Reject, as it requires a rewrite to fit in 3 page limit.

---

### Decision · Program_Chairs · 2021-05-11

Reject